# Structure-function analysis of the cyclic β-1,2-glucan synthase from *Agrobacterium tumefaciens*

Jaroslaw Sedzicki [1,3], Dongchun Ni [2,3], Frank Lehmann[1], Henning Stahlberg [2] ✉ & Christoph Dehio [1] ✉

The synthesis of complex sugars is a key aspect of microbial biology. Cyclic β-1,2-glucan (CβG) is a circular polysaccharide critical for host interactions of many bacteria, including major pathogens of humans (*Brucella*) and plants (*Agrobacterium*). CβG is produced by the cyclic glucan synthase (Cgs), a multi-domain membrane protein. So far, its structure as well as the mechanism underlining the synthesis have not been clarified. Here we use cryo-electron microscopy (cryo-EM) and functional approaches to study Cgs from *A. tumefaciens*. We determine the structure of this complex protein machinery and clarify key aspects of CβG synthesis, revealing a distinct mechanism that uses a tyrosine-linked oligosaccharide intermediate in cycles of polymerization and processing of the glucan chain. Our research opens possibilities for combating pathogens that rely on polysaccharide virulence factors and may lead to synthetic biology approaches for producing complex cyclic sugars.

Synthesis of different types of polysaccharides is critical for the biology of microorganisms. They constitute integral parts of the bacterial cell (e.g., peptidoglycan), are exposed on the bacterial surface (e.g., lipopolysaccharide, capsule components) or are secreted to the environment (e.g., glycans constituting the matrix of biofilms). Many polysaccharide virulence factors allow bacteria to invade their hosts and persist in them.

Cyclic glucans are produced by a number of Gram-negative bacteria[1]. Their structure, properties and function vary. In *Enterobacteria*, the cyclic enterobacterial common antigen plays a role in maintaining the outer membrane permeability barrier[2]. In *Pseudomonas*, cyclic β-1,3-glucans were shown to sequester antimicrobial molecules within biofilms, leading to increased antibiotic tolerance[3,4].

In *Rhizobiales*, cyclic β-1,2-glucans (CβG) are required for the proper host colonization. *Rhizobiales* include a number of pathogens and symbionts characterized by complex interactions with a broad range of hosts. Examples vary from human infection by zoonotic pathogens from the genus *Brucella*[5,6], to plant root colonization by pathogenic (*Agrobacterium*) and symbiotic (*Rhizobium*) species[7]. CβGs

have been implicated in osmotic adaptation, host environment recognition, immune modulation and lipid raft remodeling[8]. Insights into the synthesis could have a large impact for understanding pathogenicity.

The production of the main CβG chain occurs in the cytoplasm and is orchestrated by the cyclic glucan synthase (Cgs), a large, multi-domain membrane protein that contains several enzymatic sites (Fig. 1a). The mechanism driving CβG synthesis has only been partially deciphered, and many key aspects are not understood. The following steps have been proposed: (1) initiation involving the autoglycosylation of Cgs; (2) elongation of the glucan chain by the addition of Glc from UDP-Glc; (3) length control of the sugar chain performed by the C-terminal phosphorylase domain; and (4) cyclization that requires an intramolecular transglycosylation reaction[9–12]. The determination of Cgs structure and the interactions between different domains is critical for providing a comprehensive model. The mechanistic understanding of CβG synthesis can lead to strategies of controlling microbes that rely on polysaccharide virulence factors. From the synthetic biology perspective, it may result

[1]Biozentrum, University of Basel, Basel CH-4056, Switzerland. [2]Laboratory of Biological Electron Microscopy (LBEM), IPHYS, SB, EPFL, and Department of Fundamental Microbiology, Faculty of Biology and Medicine, University of Lausanne, Lausanne CH-1015, Switzerland. [3]These authors contributed equally: Jaroslaw Sedzicki, Dongchun Ni. ✉e-mail: henning.stahlberg@epfl.ch; christoph.dehio@unibas.ch

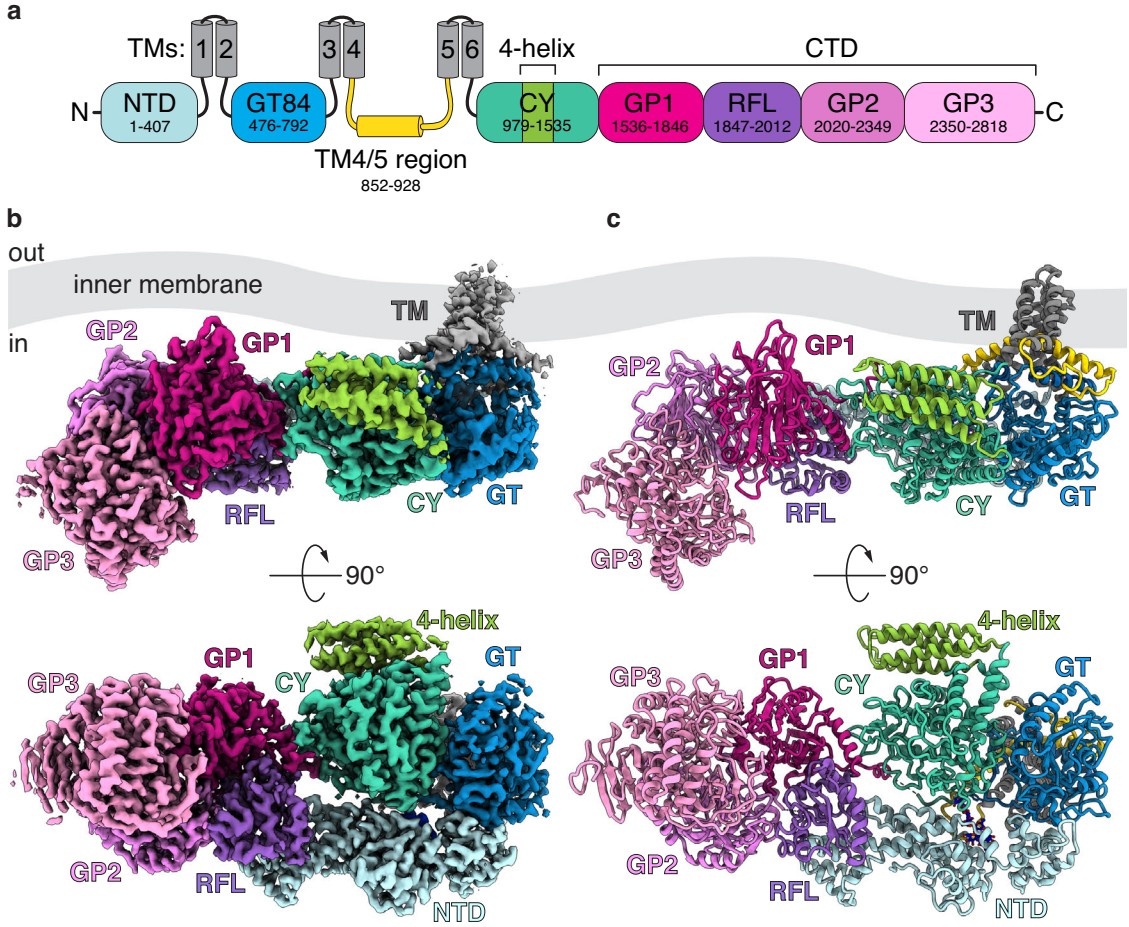

**Fig. 1 | Overview of the *A. tumefaciens* Cgs structure. a** Domain topology of Cgs. Cgs is embedded in the inner membrane with six TM helices (TMs 1–6, gray). All soluble domains are located on the cytoplasmic side of the membrane. They include the N-terminal domain (NTD, pale blue), a GT84 glycosyltransferase domain (GT, blue), putative cyclase domain (CY, teal) and a GH94 family phosphorylase domain (GP subdomains 1–3), with an additional Rossmann fold-like subdomain (RFL, purple) inserted between GP1 and GP2. In addition, there is an

extended stretch of 76 residues between TM helices 4 and 5 (yellow). The range of residues corresponding to different domains is indicated. Cryo-EM map (**b**) and model (**c**) of the full-length Cgs (Cgs$_{iGP1}$ model). Side (top) and cytoplasmic (bottom) views are shown. Coloring scheme same as (**a**). The protein can be divided into two major parts: the NTD/GT/CY region located in the proximity of the TMs, and a rigid C-terminal GP. The region between TMs 4 and 5 (yellow) forms a helix/loop motif at the TM-GT interface.

in approaches for generating complex sugars with desired properties.

In this study, we present a series of cryo-electron microscopy (cryo-EM) structures of full-length Cgs from *A. tumefaciens* (*Atu*) in conformational states relevant to its physiological function. We were able to characterize the architecture of this complex, multi-domain glucan synthesis platform and provide insights into its enzymatic activities and regulation. Our results reveal a distinct mechanism that uses a tyrosine-linked oligosaccharide as an intermediate in cycles of polymerization and processing of the glucan chain. Our work improves the understanding of complex polysaccharide synthesis and can potentially provide a basis for approaches to fighting pathogens that rely on cyclic glucans.

## Results

### Cryo-EM analysis reveals the full-length structure of Cgs

According to previous studies[9–12], CβG synthesis involves the cooperative action of two enzymatic domains performing chain elongations and length control. These are a GT84-family glycosyltransferase (GT) and a C-terminal GH94-family phosphorylase (GP). The GT domain is believed to additionally initiate glucan synthesis by autoglycosylating Cgs at an unknown residue[12]. A putative third domain, referred to as

the "cyclase" (CY), has been implied in the cyclization of the glucan chain, but its identity remains unknown (Fig. 1a).

In order to determine the structure of Cgs, we constructed a strain of *A. tumefaciens* C58 carrying an insertion in the chromosomal copy of the *chvB* gene encoding a C-terminal 3xFlag-tag (Cgs$_{WT}$). In addition, catalytically inactive mutants of the GT (Cgs$_{iGT}$) and GP (Cgs$_{iGP}$) domains were generated by introducing substitutions of the predicted active site residues (D624A/D626A/D739A and D2393A/D2528A, respectively).

The resulting three constructs were purified, reconstituted into lipid nanodiscs[13] and analyzed by cryo-EM in the presence or absence of UDP-glucose, leading to a number of distinct density maps (for an overview, see Supplementary Fig. 1, Supplementary Figs. 7–10). The Cgs$_{WT}$ and Cgs$_{iGP}$ samples produced high resolution maps of the full-length protein. The Cgs$_{iGP}$ mutant displays reduced flexibility compared to Cgs$_{WT}$, resulting in improved cryo-EM maps obtained from smaller datasets: Cgs$_{iGP1}$ for the APO dataset, Cgs$_{iGP2}$ for the UDP-Glc dataset, and Cgs$_{iGP3}$ for a subset of the UDP-Glc dataset. All maps display a similar domain arrangement, with differences in non-protein densities (UDP-Glc molecules, glucan chain intermediates) at different sites. These likely represent distinct steps of the CβG synthesis cycle. In addition, a subpopulation of dimers was observed in the Cgs$_{iGP}$ APO

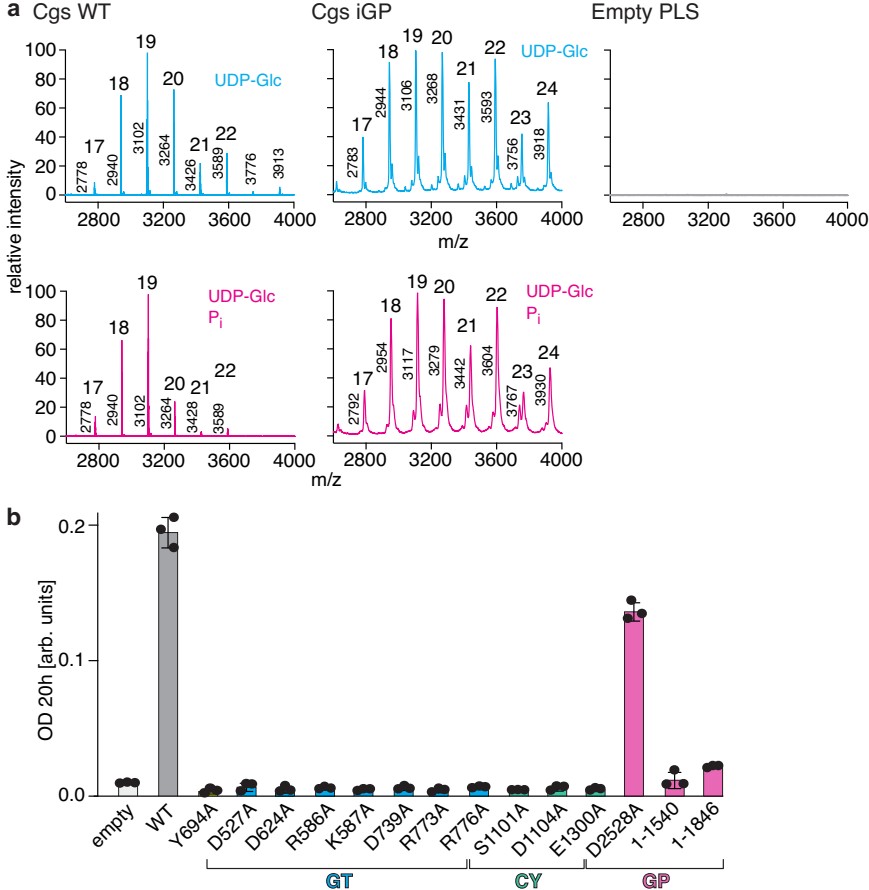

**Fig. 2 | Activity readouts of different Cgs constructs. a** In vitro CβG synthesis by Cgs proteoliposomes. MALDI-TOF analysis of reaction products reveals the characteristic size distribution of CβG species. Addition of phosphate in the sample buffer (magenta) results in the synthesis of shorter glucan chains, indicating that the length control depends on the activity of the GP phosphorylase. The number of Glc molecules within the glucan chain and the measured molecular weights are indicated. **b** Hypoosmotic growth assay showing the role of different domains of Cgs in its function. The Δcgs mutant was complemented by plasmid-encoded cgs variants. Substitutions of key residues of the GT and CY domains abolish Cgs activity. While the catalytic residue of the GP domain is not essential, truncation of the entire CTD (Cgs_{1-1587}) or its part (Cgs_{1-1846}) leads to loss of activity. Error bars represent standard deviation (SD) of three biological replicates ($n = 3$). Source data are provided as a Source Data file.

dataset, allowing us to obtain the low-resolution Cgs_{iGP4} map (Supplementary Fig. 8). Finally, the Cgs_{iGT} mutations resulted in a partially disordered protein that only gave rise to a low-resolution map.

## Domain architecture of Cgs

Our data reveal a multi-domain structure embedded in the membrane through a small, 6-helix TM domain. The protein has an elongated shape that extends perpendicular to the TM domain, suggesting that in the native membrane-embedded state it is aligned along the lipid bilayer. The soluble part of the protein is divided into 4 distinct domains (NTD, GT, CY and GP), all located on the cytoplasmic side (Fig. 1a–c).

The N-terminal domain (NTD) is characterized by an elongated fold that forms a side scaffold interacting with all the remaining parts of the protein. The GT84 domain is docked directly below the TM helices, resembling to GT2 family glycosyltransferases[14]. Residues 979–1535, corresponding to the putative CY, form a globular domain that sits at the center of the structure. The C-terminal half of Cgs (GP, starting with residue 1536) forms a rigid structure that can be divided into multiple subdomains, and resembles the GH94 family 1,2-β-oligoglucan phosphorylase from *L. phytofermentans* (LpSOGP)[15] (Fig. 1a–c, Supplementary Fig. 6a, b). In addition to three subdomains found in LpSOGP (GP1-3), Cgs has a small Rossmann fold-like subdomain (RFL, residues 1847–2012)[16] inserted between GP1 and GP2, which interacts with the NTD.

## The CY domain is homologous to GH144 family endoglucanases

The fold of the CY domain indicates homology the GH144 family endoglucanase CpSGL from *C. pinensis*[17] (Fig. 1b, c, Supplementary Fig. 2a, b), which displays activity towards β-1,2-glucans. The nature of GH144 catalytic site residues is unclear, and the enzymes were proposed to follow a non-canonical reaction mechanism. Out of three acidic residues shown to play a role in CpSGL activity (D139, E142 and E211), two have structural homologs in Cgs (D1104 and E1300 corresponding to E142 and E211, respectively). The CpSGL D139, however, is replaced by S1101 in the CY domain. The Cgs_{WT} map contains a disordered density docked at the active site of CY, which could potentially correspond to a glucan chain reaction intermediate (Supplementary Fig. 2a, b).

In addition to the GH144 fold, CY contains a stretch of around 120 residues, which forms a 4-helix motif protruding to the outside of the structure. The Cgs_{iGP4} map shows that this motif participates in the formation of a back-to-back dimer through an interaction with the CTD from a second Cgs molecule (Supplementary Fig. 2a, c). The dimer constitutes a subpopulation of particles in the datasets. Interestingly, the TM domains of the dimerized Cgs molecules are at an angle of ~90°

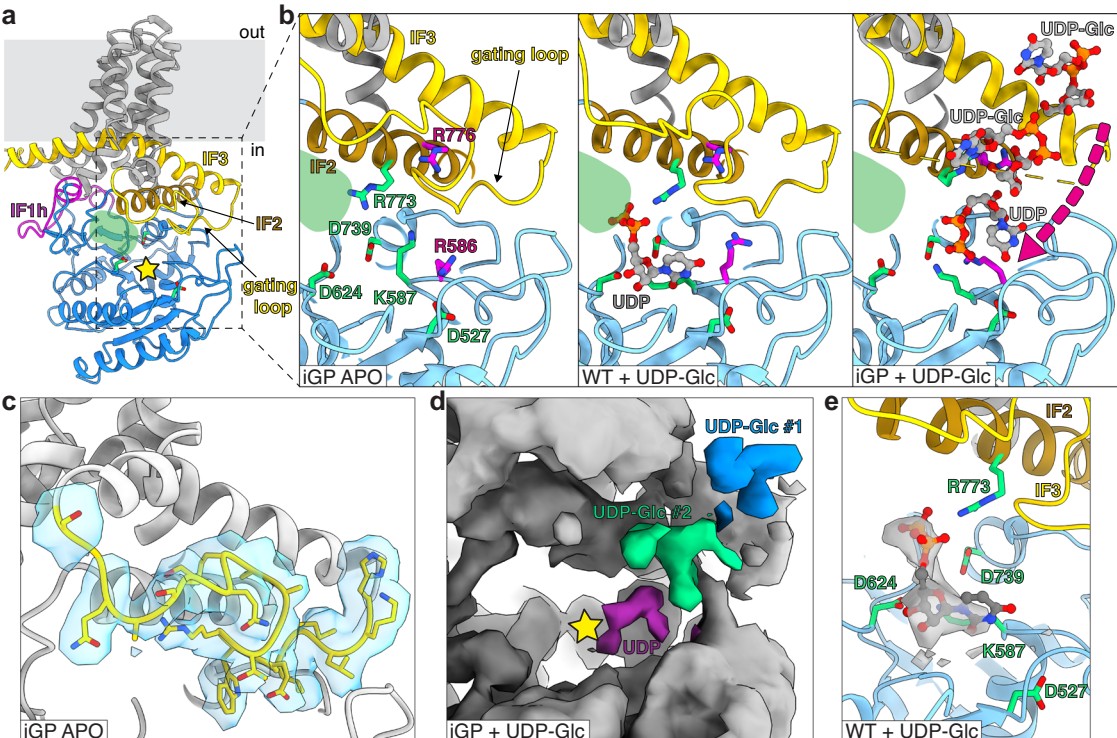

**Fig. 3 | The glycosyltransferase domain. a** The GT domain (blue) is located below a compact TM domain (gray). IF1-homology (IF1h, magenta), IF2 (brown) and IF3 (yellow) motifs are located at the interface. The modification of the IF1 motif allows access to the other side of the domain through a channel underneath of the TM bundle (green shading). The GT active site is marked with a star. The NTD and CY domains are not shown for clarity. **b** Close-up of the GT active site of three models: Cgs$_{iGP1}$ (left), Cgs$_{WT}$ (middle) Cgs$_{iGP2}$ (right). A number of residues conserved between Cgs and GT-2 family enzymes are indicated in green. The Cgs$_{WT}$ model contains an additional density docked at the active site that has been assigned as a UDP molecule. In the presence of the substrate, the gating loop of Cgs$_{iGP2}$ is disordered and additional small densities can be found near the IF2/IF3 motifs, representing a putative substrate-binding site. Residues R586, R776 (magenta) located in proximity, potentially participate in the binding. **c** Close up of the resting-state gating loop (yellow) in the Cgs$_{iGP1}$ model and the corresponding density map (blue). **d** Close up of the GT active site Cgs$_{iGP2}$ map, with the additional densities indicated in blue, green and purple. The UDP (purple) was found near the GT active site. **e** Close up of the GT active site in the Cgs$_{WT}$ model. The putative UDP density in shown in gray.

relative to each other, indicating that the interaction could introduce curvature to the aligned inner membrane.

### Significance of the various Cgs domains in CβG synthesis

Previously reported Cgs activity assays were based on whole bacteria membrane fractions, and it has been unclear if the enzyme alone is sufficient for synthesizing CβGs. To exclude the role of other, unknown factors, we reconstituted purified Cgs into proteoliposomes (PLS) and performed an in vitro activity assay. MALDI-TOF analysis of soluble fractions confirmed that Cgs PLS can convert UDP-Glc into CβG (Fig. 2a, Supplementary Fig. 3). The addition of phosphate ions resulted in shorter glucan species, confirming that the length control depends on the phosphorylase activity of the GP domain[11,15].

We wanted to further test the importance of the three enzymatic domains of Cgs for the function. To this end, we used the hypoosmotic growth assay, which is based on the sensitivity of CβG-deficient *Atu* to hypoosmotic stress (Fig. 2b, Supplementary Fig. 3d–g)[18]. A Δ*chvB* mutant of *Atu* was generated and different constructs of the *chvB* gene were tested for their ability to rescue the deletion phenotype. Multiple enzymatic pocket residues in the GT and CY domains, which were previously shown to play a role in the activity of their respective homologs, are required for growth, indicating a key role of both domains in CβG synthesis. Inactivation of the GP enzymatic activity by replacement of the catalytic acid residue (D2528A)[15] had only a minor impact on bacterial survival, suggesting that glucan length control is not critical for Cgs functionality, which is in accordance with previous studies[11]. The presence of the CTD domain itself, however, seems

important, and its complete (Cgs$_{1-1540}$) or partial truncation (Cgs$_{1-1846}$) has a detrimental effect. This hints to a scaffolding role of the CTD that is independent of length-control.

### The GT84 domain displays a modified GT2 family fold

The GT84 domain responsible for the synthesis of the linear glucan chain is located directly below the transmembrane domain (Fig. 3a, b, Supplementary Fig. 4). The structure of the domain and the arrangement relative to the TM helices resembles membrane-embedded family-2 GTs as exemplified by the cellulose synthase BcsA (Supplementary Fig. 4a–d)[14]. Characteristic conserved motifs can be identified, including the D527/D624 that coordinate the UDP, the catalytic base D739, as well as the R773 of the RW motif (BcsA D179/D246, D343 and R382, respectively). Models Cgs$_{iGP1}$ and Cgs$_{iGP2}$ indicate that the presence of UDP-Glc affects the gating loop conformation (Fig. 3b–d). In the apo state, the loop can be found above the GT active site. In the presence of the substrate, the loop seems disordered, and there are additional densities bound in the proximity of IF2 and IF3. These densities could potentially indicate the binding site of UDP-Glc, although the resolution is insufficient to unambiguously assign their identity. Substitution of two arginine residues (R586 and R776) in the proximity had a strong influence on Cgs activity (Fig. 2b). It is possible that the substrate binding induces conformational changes in the gating loop and modulates the GT activity. This could indicate a potential binding site involved in allosteric regulation or recruitment of UDP-Glc to the active site.

Major differences between Cgs and BcsA can be observed. In Cgs, the TM domain is relatively compact and does not form an export

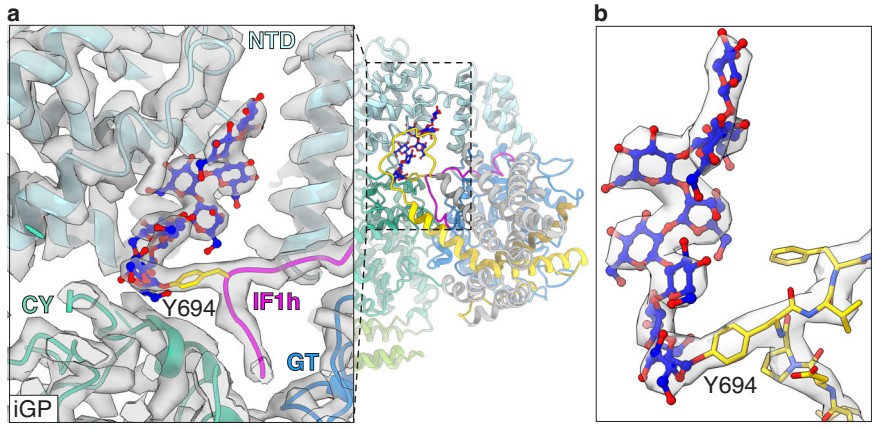

**c** Cgs WT

| Peptide sequence | Obs. Mass | ΔMass | Predicted modification |
|---|---|---|---|
| GIDPYVFTVSDVYQDLTSEGTFTGK | 4249.679 | 1511.383 | Unknown |
| GIDPYVFTVSDVYQDLTSEGTFTGK | 4196.771 | 1458.474 | 9-Hexose chain |
| GIDPYVFTVSDVYQDLTSEGTFTGK | 3224.454 | 486.158 | 3-Hexose chain |

Cgs iGT

| Peptide sequence | Obs. Mass | ΔMass | Predicted modification |
|---|---|---|---|
| GIDPYVFTVSDVYQDLTSEGTFTGK | 2791.205 | 52.909 | Replacement of $3H^+$ by $Fe^{3+}$ |
| GIDPYVFTVSDVYQDLTSEGTFTGK | 2755.318 | 17.022 | Replacement of $H^+$ by $NH_4^+$ |
| GIDPYVFTVSDVYQDLTSEGTFTGK | 2776.237 | 37.941 | Replacement of $2H^+$ by $Ca^{2+}$ |
| GIDPYVFTVSDVYQDLTSEGTFTGK | 2776.244 | 37.948 | Replacement of $2H^+$ by $Ca^{2+}$ |
| GIDPYVFTVSDVYQDLTSEGTFTGK | 2791.206 | 52.909 | Replacement of $3H^+$ by $Fe^{3+}$ |
| GIDPYVFTVSDVYQDLTSEGTFTGK | 2780.307 | 42.011 | Acetylation |

**Fig. 4 | Cgs is O-glycosylated at the Y694 residue of the IF1h loop. a** $Cgs_{iGP1}$ model seen from the side of the membrane. The glucan primer chamber is formed between the NTD (light blue), GT (blue) and CY (teal) domains. The Y694 residue at the tip of the IF1-homology loop (magenta) is inserted into the chamber. The additional density originating form Y694 indicates a linear O-glycan chain (blue-red sticks) attached to the protein. The feature is best resolved in the maps obtained for the iGP mutant. **b** Close-up of the IF1h loop (yellow) and glucan primer (blue-red) with corresponding density map (gray). **c** Mass spectroscopy results of purified $Cgs_{WT}$ and $Cgs_{iGT}$. Peptides containing the Y694 glycosylation site were detected. The table list the observed masses (Obs. Mass) as well as the difference between the observed mass and the theoretical mass of the peptide (ΔMass). Modifications corresponding to 9- and 3-hexose chains were detected for $Cgs_{WT}$, but were absent in the $Cgs_{iGT}$ sample. Contour levels are 1.2.

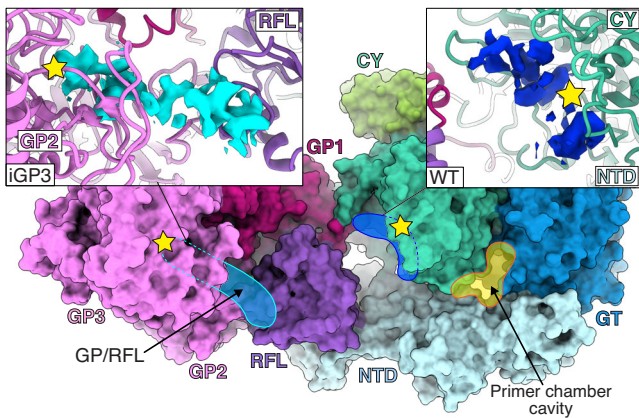

**Fig. 5 | The CTD forms a glucan-binding landscape for glucan processing.** Cytoplasmic view of Cgs. The glucan-binding landscape constitutes a large surface between the active sites (yellow stars) of the CY (teal) and GP (pink). It contains putative affinity sites for the glucan (blue shading). The landscape allows the coordination of the glucan chain during the processing by the CY and GP domains. Side panels indicate potential glucan densities that were observed interacting with the surface in different maps: at the CY active site ($Cgs_{WT}$ map, right), at the GP2/RFL interface, ending at the GP active site ($Cgs_{iGP3}$ map, left).

channel. While the interface helices 2 and 3 (IF2 and IF3) and the gating loop are present, the region homologous to IF1 (IF1h, residues 684-701) forms a loop that extends into a chamber at the interface between the NTD and CY domains. This rearrangement opens a channel from the

GT active site that continues underneath the TM domain towards the NTD (Fig. 3a, Supplementary Fig. 4a, d). The channel does not contain a density corresponding to the linear glucan chain in any of the maps. The PilZ domain, responsible for binding of the second messenger cyclic-di-GMP and regulation of the gating loop in BcsA, is absent in Cgs (Supplementary Fig. 4a).

**Cgs autoglycosylates the IF1h loop to form the glucan primer**

It was proposed that a short, covalently-linked glucan chain remains permanently associated to the protein and serves as a primer for elongation during glucan synthesis. Although the GT84 domain was proposed to autoglycosylate Cgs, the mechanism as well as the site of this modification have been unclear[12]. Our cryo-EM maps contain a polysaccharide density attached to the residue Y694, which is a part of the IF1-homology loop (Fig. 4, Supplementary Fig. 5). The density corresponds to a chain of minimum 8–9 glucose molecules that interacts with the tetratricopeptide (TPR) homology motif of the NTD. The IF1h loop and the O-glucan are located in a large chamber formed at the interface between the NTD, GT and CY domains, which is accessible from the cytoplasm through a large cavity (Fig. 5, Supplementary Fig. 5c). The Y694A mutation abolishes growth in the osmotic stress assay, indicating the importance of the glycosylation site (Fig. 2b).

While O-glycosylation of Ser and Thr residues is well studied, there is sparse information regarding Tyr[19–21]. One example is glycogenin, which is capable of both catalyzing Tyr O-glycosylation as well as extension of a short glucan chain[19]. To confirm that the GT domain is O-glycosylating Y694, we used mass spectroscopy to assess the modification state of the residue in purified $Cgs_{WT}$ and $Cgs_{iGT}$ (Fig. 4c). We found that an additional mass corresponding to 9- and 3-glucose

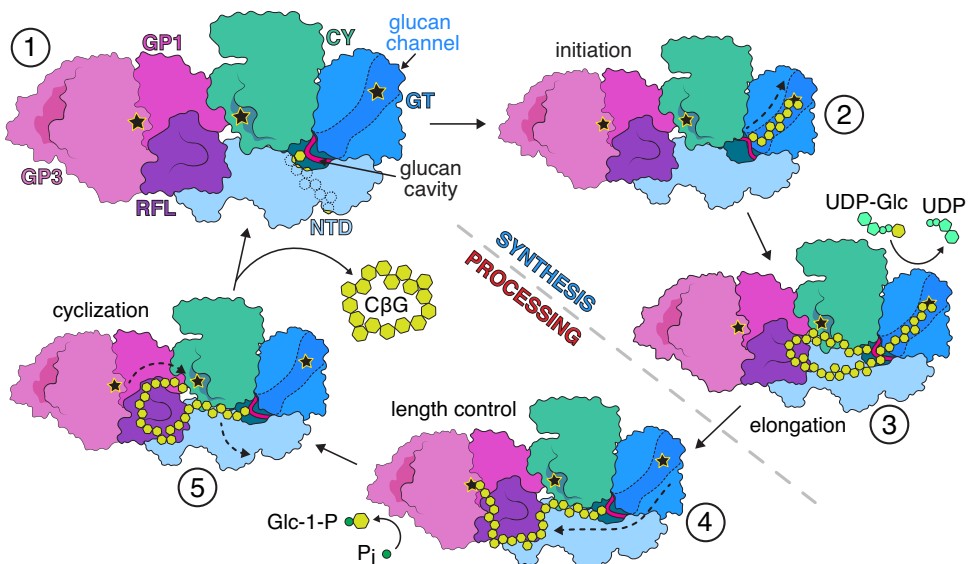

**Fig. 6 | Proposed mechanism of cyclic glucan synthesis by Cgs.** The Cgs cycle can be separated into two major stages: glucan synthesis (driven by the GT domain) and processing (driven by both CY and GP). Coloring of individual domains is indicated. Active sites of enzymatic domains are marked with stars. In the resting state (1), the primer resides in the primer chamber and is interacting with the NTD. The synthesis requires a number of rearrangements of the glucan chain (black dashed arrows). (2) The synthesis requires the entry of the non-reducing end of the primer into the GT active site. The elongation of the chain (3) continues until its length allows contact with the RFL surface (4), which leads to the translocation of the non-reducing end of the glucan and length control. (5) The RFL interaction orients the glucan chain in a conformation that promotes cyclization, leading to the release of a cyclic glucan molecule. The reaction regenerates the primer, which can be re-used in a subsequent cycle. Dashed gray line separates the synthesis and processing stages of the cycle. Dashed arrows indicate large glucan chain rearrangements. Stars indicate catalytic pockets. See main text for a detailed description.

chains is attached to the IF1-homology loop in $Cgs_{WT}$, while no such modifications were detected in $Cgs_{iGT}$.

While our results clearly indicate the autoglycosylation of Cgs, Y694 is located relatively far from the GT active site. The low-resolution $Cgs_{iGT}$ map indicates that in the absence of the O-glucan chain, the interface between the NTD and GT is disrupted, resulting in increased flexibility (Supplementary Fig. 5d). It is possible that this allows the access of the IF1h loop to the active site in newly-synthesized Cgs molecules, enabling glucan primer synthesis that leads to the mature conformation of the protein.

### The CTD forms a glucan binding landscape

The GP domain of Cgs constitutes a large, rigid fold (Fig. 5, Supplementary Fig. 6a, b). Three of its subdomains bear close resemblance to the GH94 family phosphorylase[15] (Supplementary Fig. 6b). Domains GP1 and GP2 are separated by an additional small Rossmann fold-like domain (RFL), with a 4-strand β-sheet sandwiched between a number of α-helices[16]. Although the GP active site is not essential for the synthesis of CβGs, truncations of the CTD have a strong effect (Fig. 2b), indicating additional functions of the domain.

The $Cgs_{WT}$ and $Cgs_{iGP3}$ maps obtained in the presence of UDP-Glc contain additional densities of glucan chain fragments bound to the protein at the cytoplasmic side of the CTD (Fig. 5). This surface, which stretches between the CY and GP enzymatic pockets, appears to contain affinity sites capable of coordinating the polysaccharide. The first site, at the CY active site ($Cgs_{WT}$ map), contains a disordered density. The second one stretches from the RFL subdomain, across GP2, towards GP3 (GP/RFL), and ends in the GP active site. We propose that this glucan binding landscape could provide a platform for ordered chain displacement, orienting the nascent polysaccharide for further processing that leads to cyclization by the CY domain.

### The N-terminal half of Cgs is highly dynamic

While the C-terminal half of Cgs is characterized by a rigid fold, the N-terminal half is highly dynamic (Supplementary Fig. 6c). In fact, most of the initially extracted particle datasets produced density maps with a well-defined CTD and weak partial density corresponding to the rest of the protein. The overall flexibility is reduced to some extent in the iGP construct. Our full-length maps likely represent relatively low energy states of the protein. The 3D variability analysis of the full-length maps indicates different kinds of interdomain movements. Most prominent ones include a hinge movement at the N- and C-terminal halves of the structure, as well as changing distance between the NTD and CY domains. In the latter case, the movement produces an opening between the primer chamber and the glucan binding landscape. This opening likely allows the passage of the sugar chain at certain steps of the synthesis cycle.

## Discussion

Cgs constitutes a self-contained, dynamic assembly line. The synthesis of CβG is a multi-stage process that requires the coordination of several enzymatic activities in time and space. Our work reveals the key parts of this machinery, including the enzymatic domains as well as motifs that allow the proper positioning and migration of the glucan chain. Our data indicate that Cgs is highly dynamic. The movement of the different parts of the protein relative to each other is apparent, especially between two halves of the molecule. This high flexibility could play a role during CβG synthesis, and there possibly are conformations important for the function that are dynamic and difficult to capture by cryo-EM.

Our work identifies the Y694 glycosylation site and provides evidence supporting the dual role of the GT domain, which facilitates both the extension of the glucan as well as the autoglycosylation that creates the glucan primer. In fully folded Cgs, access of Y694 to the active site may not be possible. At the same time, the lack of modification in the iGT mutant seems to destabilize the entire protein. In newly produced Cgs polypeptides, this increased flexibility could facilitate the addition of the O-glucan primer to Y694, leading to a synthesis-ready conformation. The autoglycosylation likely occurs

only once during the protein's lifetime and the glucan primer is reused in subsequent cycles.

The requirement for a covalent link between the glucan chain and Cgs distinguishes it from previously described membrane-embedded GTs[14,22,23]. The glucan primer can facilitate synthesis initiation and stabilize protein-sugar interactions in further steps, preventing dissociation during conformational rearrangements. Only few examples of enzymes capable of Tyr O-glycosylation are known[20,21], and only glycogenin was shown to combine this function with sugar chain extension[19]. Cgs is an example of this bifunctionality in bacteria. The substrate-induced conformational changes of the GT gating loop hint to an alternative regulatory mechanism employed in the absence of the PilZ-domain. The putative UDP-Glc binding site, which could play a role in modulating the activity of Cgs and potentially other membrane-bound family 2 GTs[22,23], needs to be validated by future studies.

Since the exact mechanism of the GH144 family hydrolases is unclear[17], the nature of the reaction catalyzed by CY also remains elusive. In particular, one of the putative active site residues proposed for CpSGL, D139, is replaced by S1101 in Cgs. One possible explanation may be that CY mediates a modified hydrolysis mechanism that leads to intramolecular transglycosylation. The exact mechanism of the reaction catalyzed by CY remains to be determined. The cyclization would require the precise coordination of different portions of the glucan chain relative to each other at the CY active site[24]. The binding landscape formed by the subdomains of the GP seems to provide an affinity platform that arranges the chain in an appropriate conformation, allowing length control and subsequent cyclization. We hypothesize that the interaction of the glucan chain with the landscape, along the distance between Y694 and the GP active site, is the first length determinant, setting the size at 17–25 Glc molecules. Additional trimming of the chain may further narrow down the size range, largely limiting the chain size to 20 subunits. Our data confirm the previous observations that length control is not essential for CβG synthesis. However, the lack of GP activity has a small impact on hypoosmotic stress tolerance. Moreover, the active site residue of GP is conserved in known Cgs homologs, indicating the importance of this enzymatic activity for fitness. It has been postulated that the absence of GP activity leads to slower glucan export and partial accumulation of the larger sugar species in the cytoplasm. This hypothesis is supported by the apparent preference of the glucan transporter Cgt for smaller glucans[18]. It is likely that the main role of length control is to provide glucans that better fit the transporter, ensuring efficient export. In addition to a direct function in CβG synthesis, both GP and CY (through the 4-helix domain) are responsible for the formation of a Cgs dimer. However, the exact role of the dimer remains unclear. The interaction could stabilize certain conformations of individual Cgs monomers or provide cues between different domains to control their sequential action.

Based on our findings, we propose a general model for CβG synthesis (Fig. 6), which can be separated into 2 major stages: synthesis (driven by GT) and processing (driven by CY and GP). The 9-glucose O-glucan primer attached to the IF1h loop is synthesized by the GT during Cgs folding and is reused in subsequent cycles. In the stable resting state, a primer is interacting with the TPR-homology part of the NTD (Fig. 6, step 1). A large cavity between the NTD, GT and CY domains allows access from the glycosylation site to the surrounding solvent. The synthesis requires the transition of the non-reducing end of the primer into the GT active site (Fig. 6, step 2). Based on structures of related GTs, this most likely occurs through the channel created by the unwinding of the IF1h. We were unable to observe the glucan chain in the channel in all obtained structures. This could be caused by a number of reasons, including the flexibility of the protein and instability of this conformational state. It is also possible that the affinity of the channel for the glucan chain is relatively low, leading to detachment during sample preparation.

The glucan primer is extended by the addition of new glucose molecules to the non-reducing end (Fig. 6, step 3). The growing sugar chain forms a loop that is pushed through the cavity. Once the chain reaches a certain critical length, it interacts with the GP, triggering glucan displacement and the transition to the processing stage. Although the exact mechanism behind the switch is difficult to deduce, it results in the coordination of the glucan chain at the binding landscape, with the non-reducing end accessible to length control at the GP active site (Fig. 6, step 4). The length control can occur at this point, but it does not seem essential for cycle completion. Finally, CY catalyzes cyclization, leading to the release of a cyclic CβG molecule and restoration of the 9-glucose primer (Fig. 6, step 5).

Future work is required to fully clarify the exact mechanism leading to the synthesis of a CβG chain. Insights into the process of CβG synthesis can provide tools to combat pathogens that use polysaccharides as virulence factors. It can also equip us with approaches for the synthesis of complex polysaccharides using biotechnology approaches.

## Methods

### Plasmids
For a detailed description of all plasmids used in this study, see Supplementary Table 3. DNA ligations were performed using the In-Fusion HD Cloning Kit (Takara Bio). The identity of all constructs and mutant strains was confirmed by sequencing (Microsynth AG). See Supplementary Table 2 for the complete list of primers and their specific purpose. The derivatives of pNPTS138 used for directed mutagenesis were generated by amplifying chromosomal DNA fragments from heat-killed *Atu* colonies and ligating them into the pNPTS138 backbone digested with SalI and HindIII. The pJS801 used in the hypoosmotic growth assay is a derivative of pBBR1 generated by amplifying the *chvB* gene sequence with the upstream promoter region from heat-killed *Atu* using primers pJS1108/prJS713. The PCR product was ligated with the pBBR1MCS-2 amplified with primers prJS751/prJS1072. The derivatives of pJS801 carrying point mutations were generated by amplifying the *chvB* gene sequence using primers introducing relevant point mutations (see Supplementary Tables 2 and 3) and re-ligating them with pBBR1.

### Bacteria strains
For a detailed description of all strains used in this study, see Supplementary Table 4. Stellar competent cells (Takara Bio) provided with the In-Fusion Cloning Kit were used in all cloning procedures. All strains were maintained in lysogeny broth (LB, 10 g l⁻¹ tryptone, 5 g l⁻¹ yeast extract, 10 g l⁻¹ NaCl). When required, kanamycin (50 μg ml⁻¹), oxytetracycline (12 μg ml⁻¹ for *E. coli* or 3 μg ml⁻¹ for *Atu*) and 5-aminolevulinic acid (50 μg ml⁻¹) were added to growth media. All *E. coli* strains were grown at 37 °C unless stated otherwise. *A. tumefaciens* C58 strains were cultivated at 28 °C. Mutagenesis vectors were introduced into *Atu* strains by conjugation using the ST18 donor strain[25]. Expression plasmids were introduced by electroporation. All chromosomal modifications of the *Atu* C58 were achieved with derivatives of the pNPTD138 suicide vector. Following conjugation into the target strain, clones that have undergone two successful recombination events were selected in a two-step process. The required genomic changes in the targeted loci were confirmed by sequencing. The *Atu* strain used in the growth assay is a derivative of *Atu* C58 with a deletion of the *cgs* gene (Δ*chvB*, *AtuOO4*) using plasmid pJS301. Rescue plasmids carrying different constructs of *cgs* were introduced into *AtuOO4* to generate strains used in the hypoosmotic growth assay. The strain used for expressing the Cgs-Flag construct was generated by introducing a genomic 3xFlag-tag sequence to the end of the *cgs* (*chvB*) gene using plasmid pJS305. The resulting *AtuOO9* strain was later used to generate two

additional expression strains carrying mutations in the active sites of GT84 (Atu130, generated using plasmid pJS091) and GH94 (Atu125, generated using plasmid pJS309). The expression of the *cgs-3xFlag* constructs in the resulting *Atu* strains was confirmed by Western blot using an anti-Flag antibody (clone M2, Sigma #F1804, 1:1000 dilution) and a secondary anti-mouse HRP conjugate antibody (Cell Signaling #7076, 1:1000 dilution).

### *Atu* growth assay

*Atu* strains were precultured overnight at 28 °C in 14 ml polypropylene tubes (Sarstedt) in a rotary shaker. The $OD_{600}$ of the cultures was measured in a BioPhotometer #6131 (Eppendorf). The cultures were then used to inoculate YPL medium (0.1% peptone, 0.1% yeast extract, 0.09% glucose) at a final $OD_{600}$ of 0.01. The cultures were then dispensed into 96-well flat bottom plate (160 µl per well) and placed Synergy H4 Hybrid Microplate Reader (BioTec) and incubated with fast shaking at 28 °C for indicated periods of time. $OD_{600}$ was measured every 60 min. Technical well duplicates were used for each strain. Each reported result represents at least 3 ($n = 3$) biological replicates.

### Cgs expression and purification

*Atu* strains carrying the chromosomal 3xFlag insertion at the C-terminus of the *chvB* gene (Atu009, Atu125, Atu130) were grown at 28 °C in LB medium until $OD_{600}$ of 4.5-5.5. The cells were collected by centrifugation and frozen at −80 °C until further processing. All purification steps were performed at 4 °C. Bacteria pellets were resuspended in lysis buffer (50 mM HEPES pH 7.5, 500 mM NaCl) supplemented with cOmplete protease inhibitor tablets (Roche) and lysed by 3 passes at 20'000 psi through a microfluidizer (Microfluidics). Large debris was removed by centrifugation (30 min, 10,000 g) and cell membranes were pelleted by ultracentrifugation (45 min, 200'000 g). Membrane pellets were solubilized with 1% n-dodecyl-β-d-maltopyranoside (DDM, Anatrace), loaded onto Flag affinity agarose beads (Sigma) and washed extensively with buffer containing 0.05% DDM. Following elution with 0.4 mg/mL 3xFlag peptide (Sigma), the samples were purified by size-exclusion chromatography on a Superose 6 Increase 10/300 column (GE Healthcare).

### Reconstitution of Cgs into nanodiscs

*E. coli* polar lipid extract (Avanti Polar Lipids) was solubilized in chloroform, dried under nitrogen gas to form a lipid film, and stored under vacuum overnight. The lipid film was resuspended at a concentration of 25 mM in buffer containing 20 mM HEPES, pH 7.5, 150 mM NaCl and 300 mM sodium cholate. Purified Cgs, the MSP1D1 membrane scaffold protein[13] and lipids were mixed at a molar ratio of 1:4:100 in buffer containing 25 mM HEPES, pH 7.5, 150 mM NaCl and incubated for 30 min. at 4 °C. Detergents were removed by incubation with 100 mg Bio-Beads SM2 (Bio-Rad) overnight at 4 °C. The Cgs-nanodisc complexes were purified using a Superose 6 Increase 10/300 column (GE Healthcare) in a buffer containing 25 mM HEPES, pH 7.5, and 150 mM NaCl.

### Reconstitution of Cgs into proteoliposomes

*E. coli* polar lipids at a concentration of 10 mg ml$^{-1}$ in 25 mM HEPES pH 7.5, 100 mM NaCl were extruded through a 100 nm filter to generate liposomes. Purified Cgs at a concentration of 2 mg ml$^{-1}$ was added to liposomes destabilized by 0.3% DDM at a 1:2'000 protein:lipid molar ration. Detergent was then removed by two rounds of fresh Bio-Beads. Resulting proteoliposomes were loaded onto a Sephadex G50 column equilibrated with buffer 1 (25 mM Tris pH 7.6, 100 mM NaCl). Concentration of protein in the eluted fractions was measured using the Bradford method (protein assay dye purchased form Bio-Rad). Fractions containing the proteoliposomes were combined and adjusted to 0.2 mg ml$^{-1}$ protein concentration. The incorporation of Cgs into proteoliposomes was confirmed by SDS-PAGE.

### CβG in vitro synthesis assay

Cgs proteoliposomes were diluted to 0.1 mg ml$^{-1}$ protein concentration in buffer 1. The final concentrations of reaction components were 0.1 mg ml$^{-1}$ protein, 5 mM UDP-Glc, 5 mM MgCl$_2$, 5 mM sodium phosphate, 5 mg ml$^{-1}$ linear β-1,2-glucan (Megazyme) in 300 µl final sample volume. After pipetting on ice, reactions were incubated for 24 h at 22 °C with slow rotation. Afterwards, the liposomes were removed by running the sample through 100 kDa cutoff Amicon centrifugal filters (Merck). Finally, the sample buffer was exchanged to water in 3 kDa cutoff Amicon centrifugal filters (Merck). The samples were then concentrated to a final volume of 30 µl and subjected to mass analysis. Mass spec analysis of glucans

### MALDI-TOF mass spectroscopy analysis

Samples were measured under conditions optimized for low mass materials (0.5–6 KDa, for the detection of glucans). 2,5-dihydroxybenzoic acid (DHB, Sigma-Aldrich) was used as the matrix compound and 50 mg/mL DHB was dissolved in a 1:500:500 v/v/v TFA/water/acetonitrile solution and then mixed with the sample in a 1:1 volume ratio. The samples were mixed. 1.5 µL of each sample was spotted on a stainless steel MALDI plate and dried in air. Mass spectrometric data were collected in reflectance mode (RP) ion mode on a MALDI-TOF/TOF mass spectrometer (Bruker AutoFlex Speed). MALDI was launched by a Nd:YAG SmartbeamTM-II 2 kHtz laser (355 nm) with 84% of intensity per spectrum, accumulated 1000 shots.

### Electron microscopy sample preparation

Purified, nanodisc-reconstituted protein samples were concentrated to 2.0–2.5 mg ml$^{-1}$. To generate substrate-bound samples, the protein was incubated with 5 mM UDP-Glc and 1.5 mM MgCl$_2$ at 22 °C for 2 h directly before freezing. 4 µl of protein solution was applied to glow-discharged Quantifoil holey carbon grids (2/2, 300 mesh). Grids were blotted for 3 s at 100% humidity and 4 °C, using a Vitrobot Mark IV (ThermoFischer).

### Cryo-electron microscopy data acquisition and processing

For the Cgs$_{WT}$ sample, cryogenic grids were screened on a Glacios TEM (TFS 200 kV) to determine the ice quality and then transferred to a Titan Krios G4 TEM (TFS 300 kV) equipped with Cold-FEG and a Falcon IV detector. EPU v2.12.1 (TFS) software was used for data automation. The range of defocus at exposure was −0.8 to −2.5 µm and the total dose calibrated was 50 e/Å2. Data were exported to EER format. For the Cgs$_{iGP}$ sample, movies were collected on a Titan Krios (TFS) operating at 300 kV, equipped with a Gatan Quantum-LS energy filter (20 eV zero-loss energy filter) and a K2 Summit direct electron detector (Gatan Corporation, California, USA). SerialEM was used for data automation[26]. Dose fractionated images were acquired in counting mode with a nominal magnification of 165 kx, corresponding to a physical pixel size of 0.82 Å. The defocus range at exposure was 1.0 to −2.2 µm. Detailed data acquisition statistics and image processing workflow are shown in Supplementary Figs. 8–10 and Supplementary Table 1. Image processing was performed with cryoSPARC v3.4[27]. Raw movies were motion corrected, dose weighted, pre-sorted and exported in mrc format. For the Cgs$_{WT}$ sample, 21,835 movies were imported into cryoSPARC v3.4. The contrast transfer function (CTF) was estimated using the cryoSPARC patch CTF implementation as well as Ctffind[27,28]. To create a 2D template, blob picking was applied to 3000 micrographs and then 2D classification was performed. Using the template picking implementation from cryoSPARC, 2,181,584 particles were then automatically populated. Due to the obviously dominant orientations of the particles, 2D rebalancing was crucial for the 2D classification process, especially with manual reduction of particles with top and bottom views. After six rounds of 2D classifications, 225,741 particles were selected and Ab-Initio & Hetero Refinement was performed, yielding an optimal 3D class with 149,348 particles.

Moreover, using cryoSPARC 3D classification, the particle subset was divided into seven classes. By combining class 4 and class 5, 49,830 particles were implemented with 3D refinement and a 3D map with a global resolution of 3.48 Å was obtained. To improve the quality of the glucan binding to the CY domain, subsequent local refinement was performed with a conventional soft mask volume surrounding the CY domain area. The final cryoEM map resolution was measured via gold standard method FSC with a cutoff value of 0.143. The local resolution distribution was estimated by the Local. Res. implementation in cryoSPARC. For the $Cgs_{iGP}$ samples, the data were processed in cryoSPARC v3.4 in a manner similar to WT samples. Data processing started with particle picking, 2D classification, ab-Initio along as well as Hetero Refinement and 3D classifications have been applied, the best 3D classes were subjected to 3D refinement and local refinement. However, a slight difference is that the flexibility and dominant orientation issues are significantly better for the $Cgs_{iGP}$ samples compared to $Cgs_{WT}$. In addition, the 2D classes of Cgs dimers could be clearly identified from the 2D classification and the dimers could be reconstructed in 3D, as shown in Supplementary Fig. 8.

## Model building and refinement
Resolutions and density qualities of the obtained cryoEM maps were sufficient for de novo model building. The initial model was built in coot v0.9.4.1 by manually tracing the backbone[29]. The sequence was then assigned and matched the densities. After several rounds of manually building and adjusting the model, the geometry, model clashes, and rotamers were optimized in Phenix v1.19.2-4158 with real_space_refine plugin[30]. The relevant statistics are in Supplementary Table 1. Figures were prepared with UCSF Chimera and UCSF ChimeraX v 1.4[31,32].

## Proteomics sample preparation
Purified $Cgs_{WT}$ and $Cgs_{iGT}$ protein samples were heated to 95 °C for 10 min. Proteins were alkylated using 15 mM iodoacetamide at 25 °C in the dark for 30 min. For each sample, 50 µg of protein lysate was captured, digested (trypsin 1/50, w/w; Promega), and desalted using STRAP cartridges (Protifi) following the manufacturer's instructions.

## LC-MS analysis of immunoprecipitated samples
Approximately 250 ng of peptides were subjected to LC–MS/MS analysis using a Q Exactive HF Mass Spectrometer fitted with an Ultimate 3000 (both Thermo Fisher Scientific) and a custom-made column heater set to 60 °C. Peptides were resolved using a RP-HPLC column (75 µm × 30 cm) packed in-house with C18 resin (ReproSil-Pur C18–AQ, 1.9 µm resin; Dr. Maisch GmbH) at a flow rate of 0.3 µL min$^{-1}$. A linear gradient of buffer B (80% acetonitrile, 0.1% formic acid in water) ranging from 7% to 30% over 30 min and from 30% to 50% over 5 min was used for peptide separation. Buffer A was 0.1% formic acid in water. The mass spectrometer was operated in DDA mode with a Top5 method. Each MS1 scan was followed by high-collision-dissociation (HCD) of the 5 most abundant precursor ions with dynamic exclusion set to 20 s. For MS1, 3e6 ions were accumulated over a maximum time of 100 ms and recorded at a resolution of 120'000 FWHM (at 200 m/z). MS2 scans were acquired at a target setting of 1e6 ions, maximum accumulation time of 200 ms and a resolution of 60'000 FWHM (at 200 m/z). Singly charged ions and ions with unassigned charge state were excluded from triggering MS2 events. The stepped normalized collision energy was set to 25, 30 and 35. The mass isolation window was set to 1.4 m/z and one microscan was acquired for each spectrum.

## Proteomics data analysis
The raw files were converted to mzML format by MSConvert in conjunction with ProteoWizard[33]. The mzML files were analyzed using FragPIpe (18.0)[34] with settings for open search[35]. In brief, the spectra were searched against a database containing the protein of interest

and common contaminants and their decoys using the following search criteria: full tryptic specificity was required (cleavage after lysine or arginine residues, unless followed by proline); 2 missed cleavages were allowed; carbamidomethylation (C) was set as fixed modification; oxidation (M), phosphorylation (STY) and acetylation (Protein N-term) were applied as variable modifications; mass tolerance of −2 m/z to 2000 m/z on precursor level.

## Reporting summary
Further information on research design is available in the Nature Portfolio Reporting Summary linked to this article.

## Data availability
All data needed to evaluate the conclusions in the paper are present in the paper and/or the Supplementary Information. PDB models generated in this study have been deposited at the Protein Data Bank with accession numbers of 8RF0 ($Cgs_{WT}$), 8RF9 ($Cgs_{iGP1}$), 8RFE ($Cgs_{iGP2}$), 8RFG ($Cgs_{iGP3}$). Cryo-EM maps determined in this study have been deposited at the Electron Microscopy Data Bank with accession numbers of EMD-19114 ($Cgs_{WT}$), EMD-19116 ($Cgs_{iGP1}$), EMD-19118 ($Cgs_{iGH2}$), EMD-19119 ($Cgs_{iGP3}$), EMD-19123 ($Cgs_{iGT}$). Raw data files and FragPipe output files related to the mass spectrometry analysis of Cgs glycosylation have been deposited at MASSIVE and are accessible via the identifier MSV000093457. Source data are provided with this paper.

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

## Acknowledgements

We thank the M. Chami, K. N. Goldie, D. Kalbermatter and C. Fernandez Rodriguez from the BioEM Lab (University of Basel) and E. Uchikawa, B. Beckert and A. Myasnikov from the Dubochet Center for Imaging Lausanne (EPFL and University of Lausanne) for the cryo-EM data collection. We thank C. Perez and G. Cebrero Acuña (University of Basel) for their help with proteoliposomes generation and handling. We thank K. Fröhlich (Proteomics Core Facility, University of Basel) for the mass spectrometry analysis. We also thank Daniel Ortiz (Mass Spectrometry and Elemental Analysis platform at EPFL) for the technical assistance provided for mass spectrometry analysis. We thank L. Siewert for help during manuscript preparation. This work was supported by the Swiss National Science Foundation (SNSF, www.snf.ch) grant 310030B_201273 (to C.D.), SNSF NCCR AntiResist grant 180541 (to C.D.), SNSF grant CRSII5_177195 (to H.S.) and Sino-Swiss Science and Technology Cooperation SSSTC grant (to H.S.).

## Author contributions

J.S., D.N and F.L. conceived the project and designed the experiments. J.S. and F.L. expressed and purified Cgs and MSP1D1 and reconstituted Cgs into lipidic nanodiscs and proteoliposomes. J.S. and D.N. performed cryo-EM data collection and processed electron microscopy data. D.N. built, refined and validated the structures. J.S. prepared DNA constructs, A. tumefaciens strains and carried out the functional assays. D.N. carried out the mass spectroscopy experiments and data analysis. J.S and D.N. prepared the manuscript. C.D. and H.S. supervised the projects and participated in manuscript writing. All authors contributed to revision of the manuscript.

## Competing interests

The authors declare no competing interests.
