## [Peer Review File · Nature Communications]

Structure-function analysis of the cyclic β -1,2-glucan synthase from *Agrobacterium tumefaciens*Editorial Note: This manuscript has been previously reviewed at another journal that is not operating a transparent peer review scheme. This document only contains reviewer comments and rebuttal letters for versions considered at *Nature Communications*. Mentions of prior referee reports have been redacted.

REVIEWER COMMENTS

Reviewer #1 (Remarks to the Author):

The paper has been revised according to my comments, and the presentation of the figures has improved significantly. Reviewer #2 should have high expertise in this work. The revision of this paper must follow his/her comments. A preprint on the structure and mechanism of *Thermoanaerobacter italicus* cyclic beta-1,2-glucan synthase appeared recently (10.1101/2023.10.30.564714), but I think the present work by Sedzicki et al. still has a high value. The authors of this preprint claim that the transglycosylase establishes a new family GHxxx other than GH144 or GH162. The CY domain in *A. tumefaciens* Cgs may be different from this "GHxxx", but just consider the possibility of creating a new family for CY. The CAZY team should be able to answer your question confidentially. I have made several comments regarding this revision.

Major points:

1. The accession numbers of EMDB and PDB are still XXX (L580-584). This is a very bad manner. I missed this problem in my previous peer review. Even in this revision, there are only unannotated validation reports with the red caution of "This wwPDB validation report is NOT for manuscript review".
2. Figure 6 legend. L361. Driven by CY and GH -> GP? The legend nicely explains the mechanism but lacks an indication for step (4). The locations of the "primer chamber" and "CTD surface" must be marked in the figure. Is the "glucan affinity sites" present in RFL? What are the yellow-lined black stars? Are they catalytic sites?
3. Extended Data Fig. 2C. The membrane is V-shaped. Is this consistent with the cellular mechanics of the bacteria?

Additional Comments:

Since reviewer #2 is not available, I have checked and will comment on the response to him/her, too. Reviewer #2 listed three points (i, ii, and iii) as unknown issues on the mechanism of Cgs. Although reviewer #2 commented "[REDACTED]", he/she appreciated that the structural basis revealed in this work is very important, especially for (i) and (ii). I agree with this. For (iii), reviewer #2 commented "[REDACTED]". However, I think this is a very strong starting point to understand the precise mechanism of Cgs although it is to be verified in the future. The authors replied to this comment adequately. The authors addressed reviewer #2's comments accurately in other replies as well. I followed up on the replies to his/her comments as below.

Major points:

1. The line numbers in the response do not match the revised manuscript. For example, the reply for point 5 indicates L223-226, but it is really L233-234.
2. L229-230. A prominent example of Tyr O-glycosylation is GH33/34 sialidases that use a Tyr residue as the catalytic nucleophile (PMID: 12812490)
3. Fig. 2a legend. "The number of Glc molecules within the glucan chain is indicated." But I see nothing in it.

4. L134. D2528 is supposed to be the catalytic acid residue, not just the catalytic residue because it corresponds to D760 in LpSOGP. D2393 corresponds to D631 in LoSOGP, and it should bind the subsite -1 Glc. This explanation must be included in the text with reference #15 (Nakajima et al., Sci. Rep., 2017).
5. I could not find the Supplemental Data of the MS data in the submitted files. This must be checked by one of the anonymous reviewers.
6. The authors removed the sequence alignment figure, which was present as Extended Fig. 1a in the previous version. The sequence alignment is important for readers with high expertise in related enzymes (GT84, GH94, GH144, and GH162) after publication. Please follow the comments given by Reviewer #2 and add residue numbers of important residues shown in the text. Reinstate this as a multi-page PDF and provide it as a supplemental figure or supplemental data.

Minor points:

1. The doi links in the reference should be <https://doi.org/10.1128/>, not <https://doi.org:10.1128/>
2. P37L702. Supplementary Table 4 (typo)
3. L249-250. the absence (of?) GP activity

Reviewer #3 (Remarks to the Author):

All my questions/suggestions have been adequately addressed. I would recommend the publication of this work providing important molecular insights into cyclic glucan synthase (Cgs).

Point-by-point response to the reviewer comments

Response to reviewer #1 and #2:

The paper has been revised according to my comments, and the presentation of the figures has improved significantly. Reviewer #2 should have high expertise in this work. The revision of this paper must follow his/her comments. A preprint on the structure and mechanism of *Thermoanaerobacter italicus* cyclic beta-1,2-glucan synthase appeared recently (10.1101/2023.10.30.564714), but I think the present work by Sedzicki et al. still has a high value. The authors of this preprint claim that the transglycosylase establishes a new family GHxxx other than GH144 or GH162. The CY domain in *A. tumefaciens* Cgs may be different from this “GHxxx”, but just consider the possibility of creating a new family for CY. The CAZY team should be able to answer your question confidentially. I have made several comments regarding this revision.

We appreciate the reviewer's recognition of the novelty and significance of our study, as well as the effort spent on revising our manuscript for the second time. Considering that the final version of the *T. italicus* manuscript will likely contain a new GH family annotation for the Cgs CY domains, we would like to refrain from establishing it in our work.

Major points:

1. The accession numbers of EMDB and PDB are still XXX (L580-584). This is a very bad manner. I missed this problem in my previous peer review. Even in this revision, there are only unannotated validation reports with the red caution of “This wwPDB validation report is NOT for manuscript review”.

The accession numbers were added to the main text. PDB reports were provided (L540-542).

2. Figure 6 legend.

- L361. Driven by CY and GH -> GP?
Corrected
- The legend nicely explains the mechanism but lacks an indication for step (4).
Step 4 was included in the figure legend.
- The locations of the “primer chamber” and “CTD surface” must be marked in the figure.
Changed ‘CTD’ to ‘RFL’ for clarity.
- Is the “glucan affinity sites” present in RFL?
Changed to ‘RFL interaction’ for clarity.
- What are the yellow-lined black stars? Are they catalytic sites?
Explanation was added to the figure legend.

3. Extended Data Fig. 2C. The membrane is V-shaped. Is this consistent with the cellular mechanics of the bacteria?

The real shape of the dimer in the bacterial cell is unknown. The purpose of the figure is to indicate the unusual architecture of the dimer and a possible influence on the lipid bilayer shape. The figure legend was adjusted to underline the hypothetical nature of this statement.

Additional Comments:

Since reviewer #2 is not available, I have checked and will comment on the response to him/her, too. Reviewer #2 listed three points (i, ii, and iii) as unknown issues on the mechanism of Cgs. Although reviewer #2 commented that “[REDACTED]”, he/she appreciated that the structural basis revealed in this work is very important, especially for (i) and (ii). I agree with this. For (iii), reviewer #2 commented that “[REDACTED]”. However, I think this is a very strong starting point to understand the precise mechanism

of Cgs although it is to be verified in the future. The authors replied to this comment adequately. The authors addressed reviewer #2's comments accurately in other replies as well. I followed up on the replies to his/her comments as below.

Major points:

1. The line numbers in the response do not match the revised manuscript. For example, the reply for point 5 indicates L223-226, but it is really L233-234.

We apologize for this. This is due to the substantial changes made to the original text.

2. L229-230. A prominent example of Tyr O-glycosylation is GH33/34 sialidases that use a Tyr residue as the catalytic nucleophile (PMID: 12812490)

Reference was added (L221).

3. Fig. 2a legend. "The number of Glc molecules within the glucan chain is indicated." But I see nothing in it.

The numbers were added in Fig. 2

4. L134. D2528 is supposed to be the catalytic acid residue, not just the catalytic residue because it corresponds to D760 in LpSOGP. D2393 corresponds to D631 in LoSOGP, and it should bind the subsite -1 Glc. This explanation must be included in the text with reference #15 (Nakajima et al., Sci. Rep., 2017).

The text was adjusted accordingly (L129)

5. I could not find the Supplemental Data of the MS data in the submitted files. This must be checked by one of the anonymous reviewers.

Raw data files and FragPipe output files related to Cgs glycosylation analysis have been deposited at MASSIVE and are accessible via the identifier MSV000093457 (password "PCF").

MS data related to in vitro cyclic glucan synthesis were added as a Supplementary Data excel sheet.

6. The authors removed the sequence alignment figure, which was present as Extended Fig. 1a in the previous version. The sequence alignment is important for readers with high expertise in related enzymes (GT84, GH94, GH144, and GH162) after publication. Please follow the comments given by Reviewer #2 and add residue numbers of important residues shown in the text. Reinstate this as a multi-page PDF and provide it as a supplemental figure or supplemental data.

A PDF with the alignments of three Cgs genes was added as a supplementary file. According to the instructions of reviewer 2, an alignment of the Cgs GP domain with the LpSOGP homolog was added.

Minor points:

1. The doi links in the reference should be <https://doi.org/10.1128/>, not <https://doi.org:10.1128/>

2. P37L702. Supplementary Table 4 (typo)

3. L249-250. the absence (of?) GP activity

Response to reviewer #3:

Reviewer #3 (Remarks to the Author):

All my questions/suggestions have been adequately addressed. I would recommend the publication of this work providing important molecular insights into cyclic glucan synthase (Cgs).

We thank the reviewer for the effort spent on revising our manuscript.

REVIEWERS' COMMENTS

Reviewer #1 (Remarks to the Author):

All my comments have been adequately addressed.